# The European tomato market. An approach by export competitiveness maps

**María de las Mercedes Capobianco-Uriarte[1], Juan Aparicio[2], Jaime De Pablo-Valenciano[1], María del Pilar Casado-Belmonte**[1] *

1 Economics and Business Department, University of Almería, Almería, Spain, 2 Center of Operations Research (CIO), University Miguel Hernandez of Elche, Alicante, Spain

* mbelmont@ual.es

**Data Availability Statement:** All relevant data are within the paper and its Supporting information files.

## Abstract

Most empirical studies examining the export competitiveness of a country in a target market are undertaken by focusing on supply, only analysing the group of competing countries. In addition, if the target market to be analysed is extensive, like the European Union, it is generally analysed as a whole. This study presents an evaluation of the tomato export competitiveness, from a differentiated demand perspective, analysing its main customers markets in the context of European Union. The methodological framework is implemented through Constant Market Share to analyze variations in exports, allowing the portion attributable to competitiveness and segregation into general or specific competitiveness to be quantified. The Constant Market Share was adapted to focus on the differentiated demand so as to observe the influence of the worldwide crisis (2007/08) on the European tomato market. This study allows the analysis of profile changes into the competitor exporting economies. As a contribution to the methodology, this study presents a new graphical way of representing the results of Constant Market Share methodology by means of export competitiveness maps in the European tomato market for the group for each main competitor in each European client market. According to our results, Spain and Belgium are candidate countries to be competitive in the main European markets.

## 1. Introduction

The panorama of the tomato commercialization must be analyzed in a turbulent agricultural market scenario due to the consumer food price crisis since 2008, as described in the Agricultural Outlook report 2014–2023, jointly developed by the Organization for Economic Co-operation and Development and the Food and Agriculture Organization [1]. In the case of the global vegetable market, Van Rijswick [2] had reported that it is still predominantly a local market and only 5% of the vegetables grown are traded internationally, nevertheless that share is increasing. He exposed that easy market access is vital for export-focused vegetable-producing countries since the most fresh vegetables are highly perishable, growing circumstances (climate, water availability), production costs, exchange rates, and trade agreements can trigger vegetable trade flows. Such is the case of the international tomato trade, which is polarized in

**Funding:** J.A. thanks the financial support from the Spanish Ministry for Economy and Competitiveness (Ministerio de Economia, Industria y Competitividad), the State Research Agency (Agencia Estatal de Investigacion) and the European Regional Development Fund (Fondo Europeo de Desarrollo Regional) under grant MTM2016-79765-P (AEI/FEDER, UE).

**Competing interests:** The authors have declared that no competing interests exist.

two of the large regional trade agreements (RTA) the North American Free Trade Agreement (NAFTA) and Customs Union & Economic Integration Agreement of the European Union (EU28). In this context, the European Union highlights as the largest exporter as well as importer market, being more open to the world tomato market than the North American market.

The importance of the tomato as the vegetable with the greatest presence in international trade and the European market as leading importer/exporter on a worldwide level of this vegetable, underline the relevance of undertaking a tomato export competitiveness study in the intra-community trade context (EU28), in order to analyze the possibilities of Spain to recover the lost leadership in 2009 against the Netherlands. Analyzing only one third of the 28 markets that make up the European Union market, more than 82% of the total European tomato market is covered.

The objectives of this article are to obtain the competitive position between competitor countries in the main European markets and to determine whether there are post- crisis profile changes into the competitor exporting economies. The Constant Market Share (CMS) analysis was applied through the supply and demand perspectives, completing the work of Capobianco et al. [3] on export competitiveness in the European tomato market. The CMS methodology was adapted to undertake a comparative analysis of export performance of intra and extra-community competitors (Netherlands, Spain, France, Belgium, Italy and Morocco) in the main European markets (German, French, British, Dutch, Italian, Swedish, Polish, Spanish and Belgian markets). A temporal analysis was undertaken for two consecutive periods (2004–2010 and 2011–2016), in order to observe the influence of the worldwide crisis on exporting economies. Furthermore, as a contribution to the methodology, the results are presented by means of competitiveness maps. The competitiveness maps, in addition to facilitating the interpretation of CMS results, provide complementary information on profile categorization, namely competitive, opportunist, non-competitive or catching-up economy. That information may be relevant for policy makers or other non-state agents, especially for the selection of competitive strategies that allow to maintain, reinforce or improve the position of their products in the target markets [4].

Thus, this study contributes to the literature in several ways. First, the novelty of the employment of CMS analysis by the impact on each individual European markets. Second, from a methodological perspective, the introduction of competitiveness maps gives information about the profile categorization of exporting economies and illustrates graphically the post crisis profile changes on the European tomato market.

The remainder of the paper unfolds as follows. Next section shows the methodological guidelines carried out for the configuration of the two-dimensional system of indicators. In addition, database of international trade chosen, identification and selection of the main supplier and consumer countries of tomatoes in the European market, election of the time periods for the post crisis analysis, a CMS background, competitiveness indexes normalization, and the Export competitiveness maps are presented. After that, the main results are shown and analyzed. Finally, the discussion and conclusion sections are offered, where the most convenient product differentiation strategies in each market are discussed and post-Brexit alternatives are commented.

## 2. Tomato market competitiveness

### 2.1. Importance of the tomato trade in the international market

The international tomato trade reached historic levels in 2016, with more than 8 [mt] valued at 8.5 bn US\$ (Fig 1). The Fideicomisos Instituidos en Relación con la Agricultura report [5]

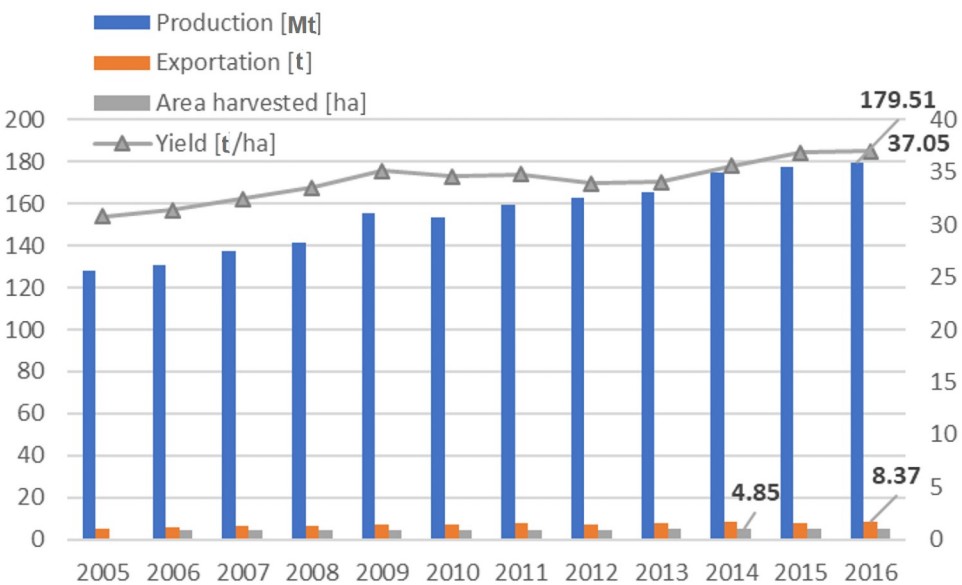

**Fig 1. Tomato world production, harvested area, yield and world exportation (2005–2016).**

indicates that the volume of world exports of tomatoes has grown at an average annual rate of 5.19% since 2007. The tomato is the most commercialised vegetable on a worldwide level, representing 20.86% (2016) of the total export volume of fresh vegetables (Fig 2) according to the United Nations Organisation data [6].

According to estimates by FAOSTAT [7], also the tomato is the most widely-cultivated vegetable in the world, reaching its historical peak in 2016 with 177.04 [mt] (Fig 1) in a total harvested area of 4.78 [mha] and with a yield of 37.02 [t/ha]. More than half of worldwide production (56.71%) is concentrated in four countries. China is the leading tomato producer worldwide (31.81%) with nearly a third of worldwide production, followed by India (10.39%), United States (7.36%), and Turkey (7.12%). The rise in tomato production between 2005 and 2016 was 29.08% at an average annual growth rate of 3.14%, slightly higher than the growth rate for worldwide horticultural production of 2.97%. The growth in production was driven both by a 13.35% (Fig 2) increase in the area harvested as well as by a 13.98% increase in yields (2005–2016). The availability of new types and varieties of vegetables and innovative methods

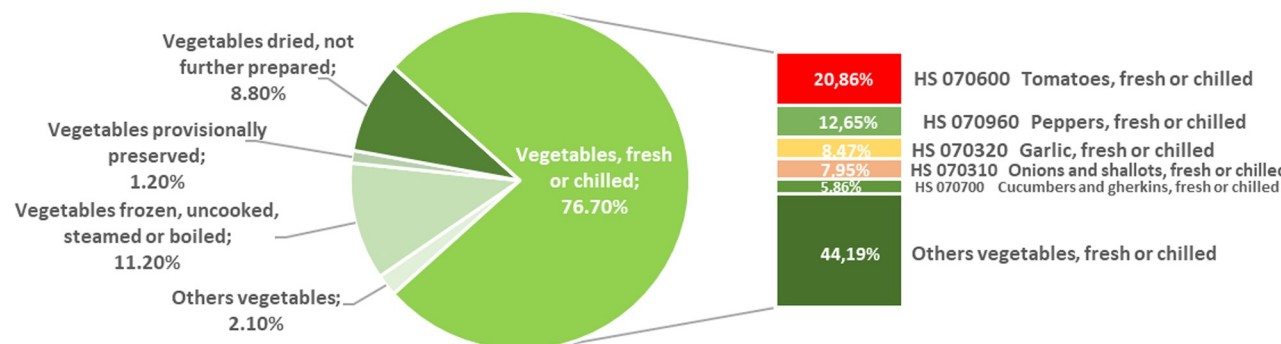

**Fig 2. World trade of fresh vegetables sector and other vegetable products (2016).**

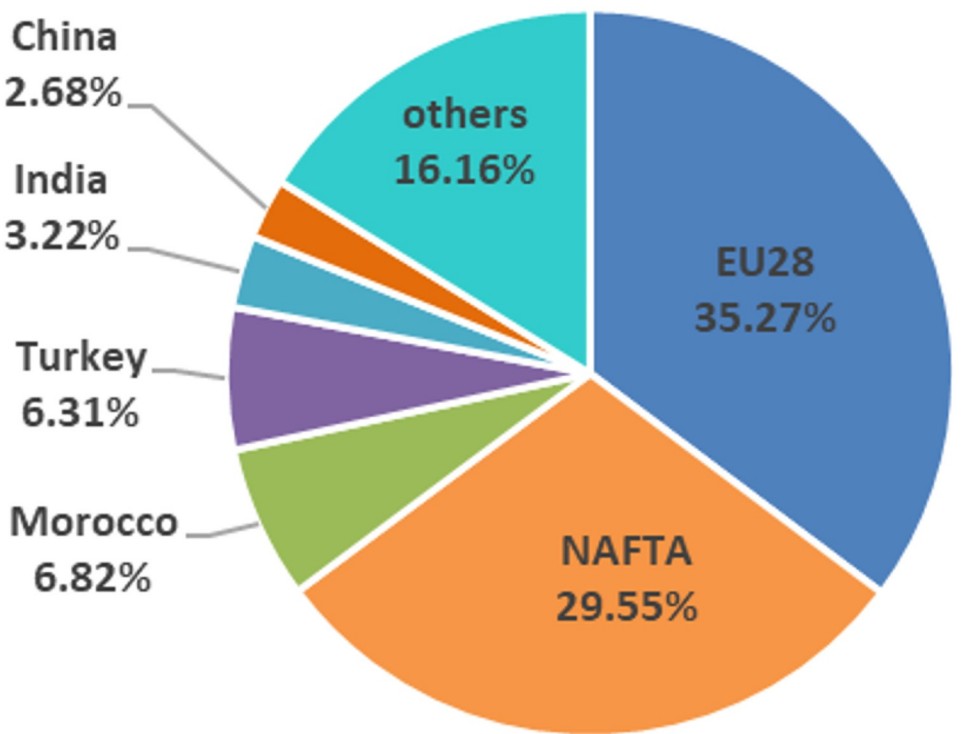

**Fig 3. Main tomatoes exporting regional commercial areas and countries (2016).**

of cultivation, together with the growing demand for vegetables, have encouraged global tomato production [5].

The main export flows in the international tomato market, between 2005 and 2016, show a stable structure since 2005 (Fig 1). The import origins and export destinations in the main commercial blocks (NAFTA and EU28) proceed or are directed mainly from member countries of RTA, characterized as intra-regional internal markets. Considering commercialization on a country level, Mexico is the world's largest exporter of tomatoes and the United States is the main importing country [5]. By contrast, if regional commercial areas are taken into account, the European Union stands out, both as the largest exporter as well as importer (Figs 3 and 4). Being the European block more open to the world tomato market than the North American market, importing more than 15% of tomatoes outside the EU and exporting about 5% of the available stock outside its frontiers (Fig 5).

Currently, the regional trade agreements that are characterized by an intense exchange of tomatoes (NAFTA and EU28) they are going through a situation of instability among their member states, causing an uncertain commercial situation on both sides of the Atlantic. The Brexit would eliminate Britain's tariff-free trade status with the other EU members and tariffs would raise the cost of exports and also increase prices of imports into the United Kingdom, where more than one third of its imports comes from the EU [8], and ranks as the third largest tomato market in the intra-Community market. The United States threat that it might withdraw from NAFTA and declaration of trade wars against China and the EU28, increase uncertainties in international trade.

It is worth noting the relevance of the tomato market in the international trade of vegetables, highlighting the European market as the leading importer/exporter. In this vein, the

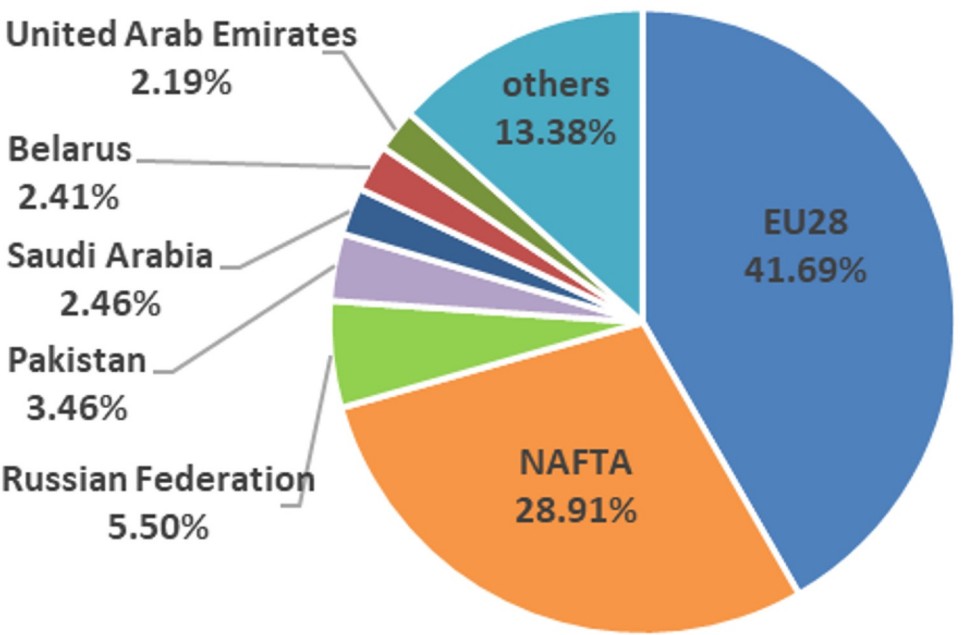

**Fig 4. Main tomatoes importing regional commercial areas and countries (2016).**

study of the tomato export competitiveness in the intra-community trade context (EU28) stands as relevant so as to find out the situation of the European tomato market (Figs 6 and 7).

In terms of the volume of production in the European Union, Italy is the largest European producer, although the area cultivated is decreasing (Fig 8) and the yield obtained is the lowest among European exporting countries in 2016 (61.94 [Tn/ha]) according to FAOSTAT data

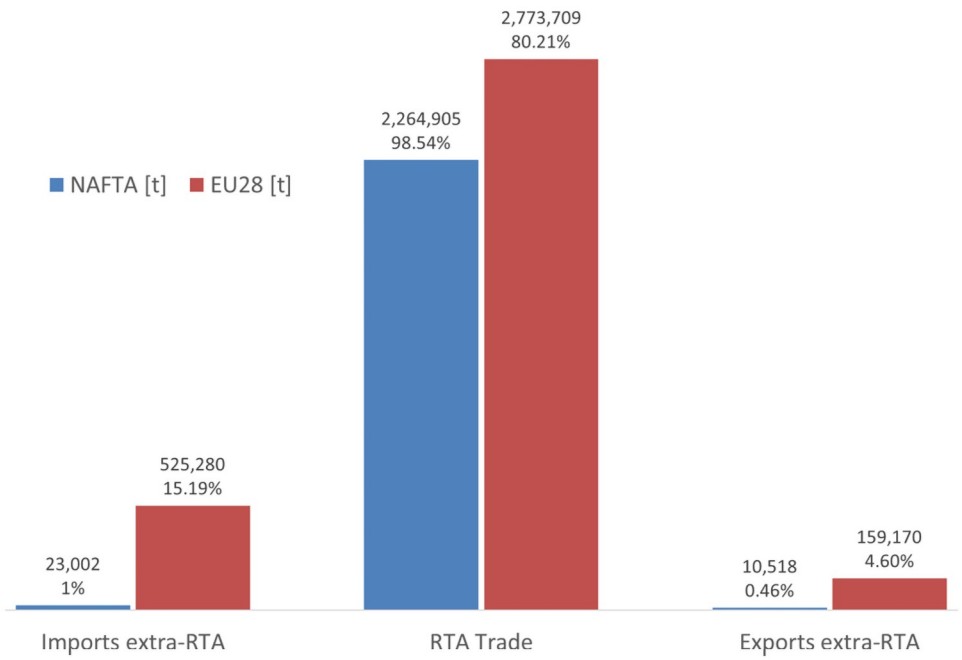

**Fig 5. NAFTA and EU28 imports, exports and intratrade of tomatoes (2016).**

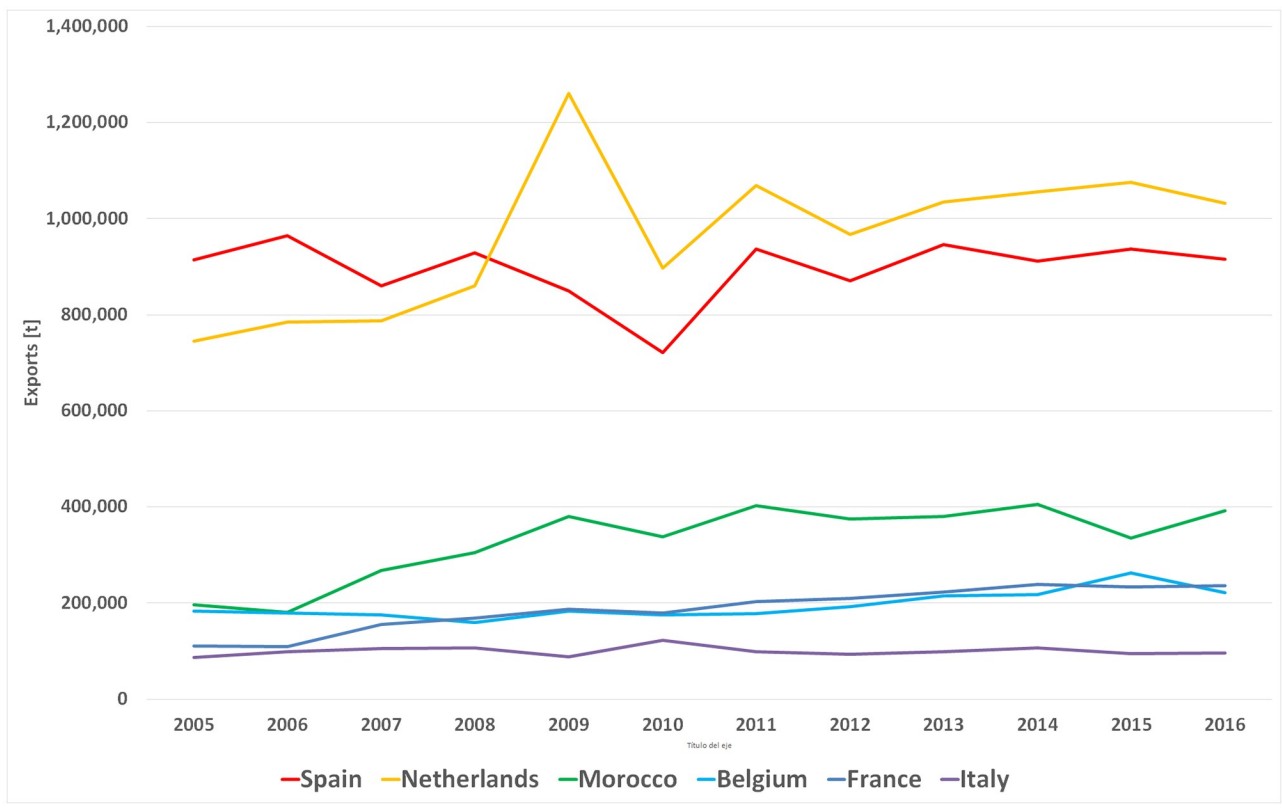

**Fig 6. Evolution of the tomatoes export volume of competitors in the European market (2005–2016).**

[7]. Spain stands out as the second largest producer, with a maximum production of 4,888 thousand tonnes in 2014. Like Italy, the area harvested has decreased over the period under consideration and has a slightly higher yield than Italy (80.80 [Tn/ha]). Spanish exports are concentrated in the months from October to May (Fig 9) and the average price is of 1.17 [USD/Kg]. The Netherlands does not stand out among the largest European producers, but shows a slight upward trend in its production, due to both an increase in the area harvested and its yield [9], presenting the highest yield in European production together with Belgium, with 506.90 [Tn/ha] in 2016. The Netherlands is also the world's second largest exporter, due to the re-export of products not produced in the Netherlands. Dutch exports are concentrated in the months April to October (Fig 9) and the average price is of 1.59 [USD/kg]. France and Belgium show the least significant values in European production. In both countries, the area harvested has decreased, while yield has increased. The average yield in France is of 186.10 [t/ha] and the average price of French and Belgian exports are around 1.41 [dollar/kg] and 1.29 [dollar/kg] respectively. Finally, Morocco's production has increased slightly, mainly due to a progressive increase in its yield (80.80 [Tn/ha]), which is similar to the Spanish yield. The average price of Moroccan exports fluctuates around 1.14 [dollars/kg].

## 2.2. Mesuaring export competitiveness

Ambastha and Momaya [10] presented a review on competitiveness theory and exposed that competitiveness is a multidimensional and unclear concept. From the beginnings of the definition of competitiveness, the concept became stratified on different levels, considering the capacity of a country to compete on an international level or to guarantee high profitability for

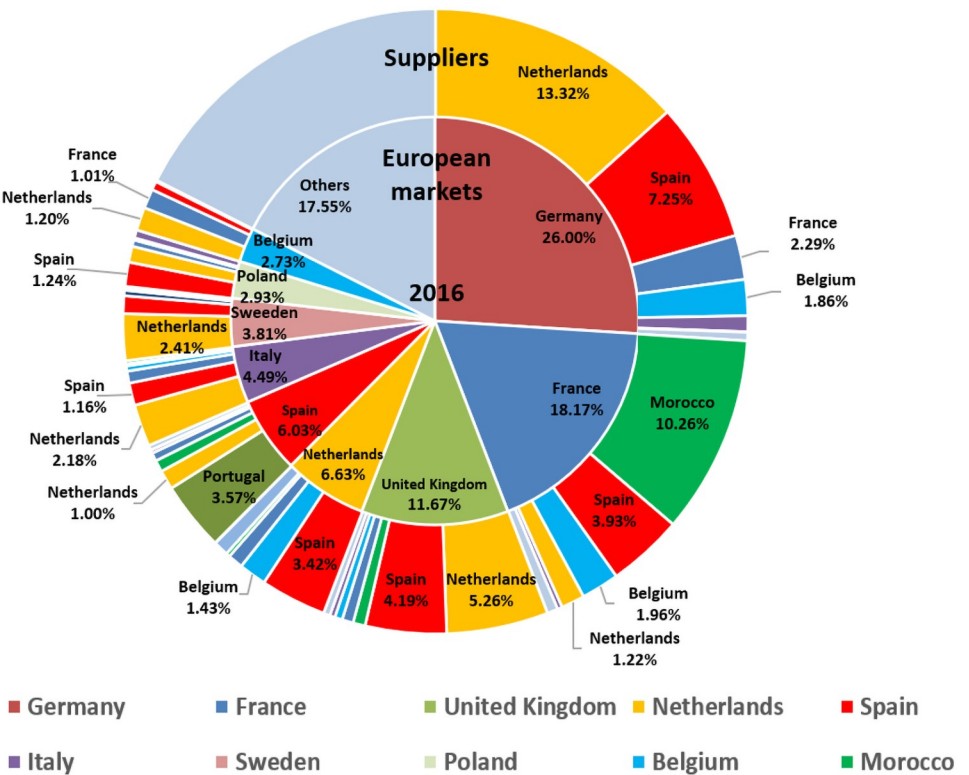

**Fig 7. Main European suppliers for importing markets of tomatoes (2016).**

the companies that make up the business fabric of a country both in its internal as well as external markets, or to gain market share on a worldwide level, or in determined natural markets or otherwise, through Regional Commercial Agreements [11]. The concept of competitiveness, according to Dussel [12], is defined as the process of dynamic integration of countries and products to international markets ex-post, depending both on the conditions of supply as well as demand. Competitiveness reflects the dynamic insertion of the products of the

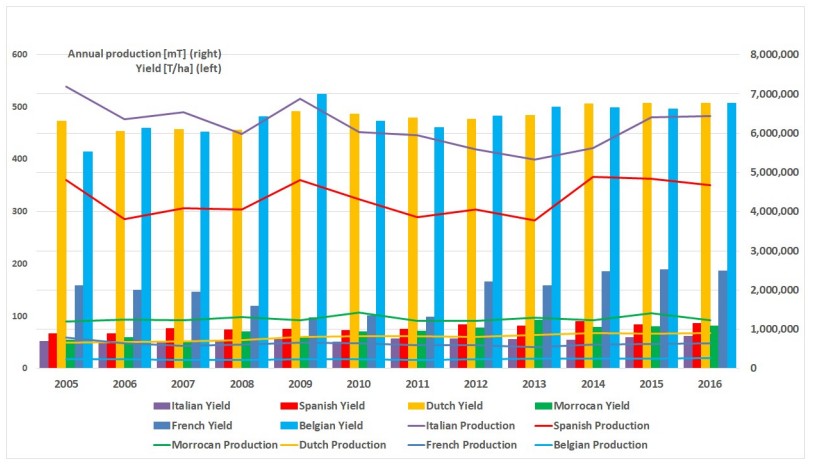

**Fig 8. Evolution of the tomato production of export competitors in the European market (2005–2016).**

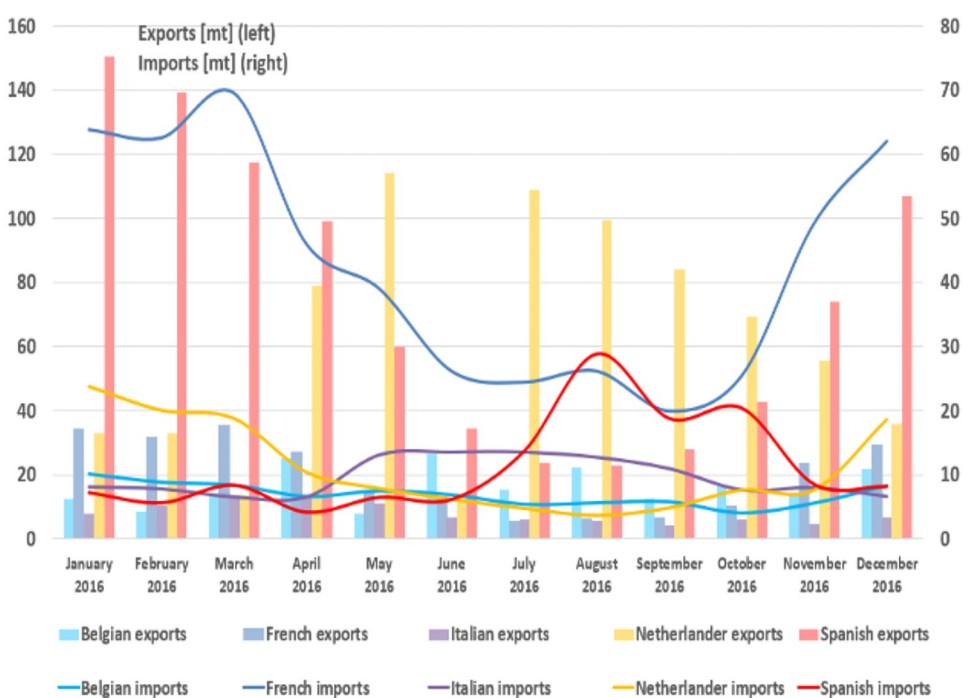

**Fig 9. Seasonal imports/exports of the main tomato re-exporters in the European market (2005–2016).**

countries selected, which depends both on the performance of the export structure compared with its competitors (supply), as well as the dynamism of international trade or specific target markets or group of clients (demand).

The tomato European trade presents a wide range of commercial roles among member states (Fig 7), some countries participate intra or extra-comunity as net exporters (Spain and Morocco) and others as net importers (Germany, the United Kingdom, Poland and Sweden), some of them share characteristics as importers-exporters (re-exporters) like France, Belgium, the Netherlands, and Italy according to seasonality (Fig 9). This wide range of commercial behaviours induces the use of more complex analyses of competitiveness, taking into account not only the analysis of the competitor group (supply perspective) in the tomatoes European market if not integrated with a differentiated analysis in each European market, especially if the target market is extensive like the European market [9].

The World Bank [13] and the European Central Bank [14] recommend the use of market share decomposition methodologies to diagnose export competitiveness. Nowadays, in the economic literature, the use of the Constant Market Share (CMS) methodology has gained importance in the last decade for the study of export competitiveness. Using data provided by the Scopus [15] and Web of Science [16] scientific literature database, the rise in publications over the last decade (2008–2018) can be observed in Fig 10. However, hitherto the CMS methodology has generally been applied to analyse the competitiveness of a group of competing countries in the European Union as target market, taking only the supply perspective into account [17–26]. To the best of our knowledge, only Molina and Taiariol's [27] work on export competitiveness has been approached from both perspectives (supply and demand), combining the competitors group analysis in the different customer markets comprising the target market. This type of differentiated-demand study is more appropriate for a target market made up of several member countries, such as the European market. In a market composed by

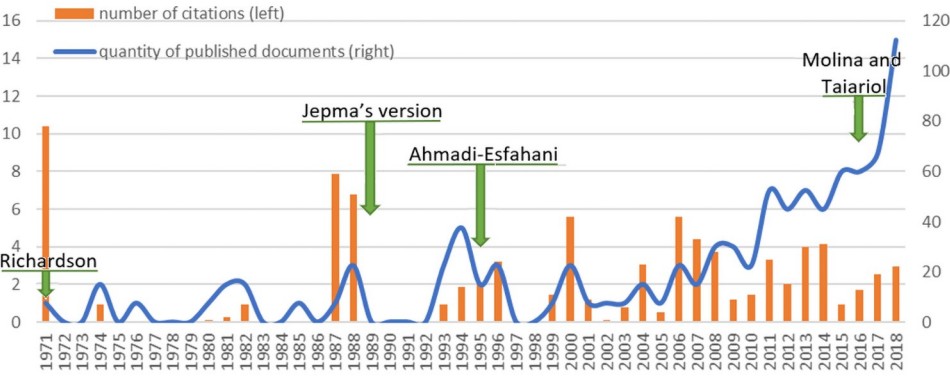

**Fig 10. Evolution of scientific production in "Constant Market Share" methodology per year (1971–2018).**

28 importing member States, if it is only analysed as a global target market, the results could mask the real situation in specific markets that comprise it.

## 3. Methodology

### 3.1. International trade database

To carry out this study, information from the statistical division of the United Nations Organization [6] was used through the harmonized system tariff code 070200 (Vegetables; tomatoes, fresh or chilled). In this work, only export data were considered, in terms of value and FOB (Free On Board), in order to avoid the bilateral asymmetry present in the official merchandise trade data.

The COMTRADE trade statistics data, like any source of information, is not free from errors and omissions and there can be multiple sources of discrepancy. The main ones are related to differences in the recording system, to registration errors, and to the exchange rate effect [28]. Differences in the recording system are due to various factors such as inclusion or exclusion of particular commodities, timing (time of recording), including valuation (imports CIF—Cost including Insurance and Freight and exports FOB). The registration errors are problems related to the treatment of low-value transactions, unregistered cross-border trade (for example Africa informal trade), missing or incomplete information (commodity classification), quantity measurement and partner country, and the intentionally incorrect reporting to avoid tariffs and quotas. Some countries do not register below a threshold (Japan, Canada, Australia, etc). Finally, the "Exchange Rate" effect consists in a distortion in the estimation of exports due to an unexpected change in the currency exchange rate that causes significant changes in the trade flows.

### 3.2. Identification and selection of the main supplier and consumer countries of tomatoes in the EU28

The wide range of commercial behaviour that characterises the markets that make up the European tomato market requires careful identification and selection of both the main exporting-supplying countries and the main importing-consuming countries. As a first selection criterion, the methodology presented by Capobianco et al. [29] was considered to identify country profiles and sub-profiles within the European tomato consumer market through a multivariate analysis of two-stage clusters. At the first level of clustering from 2005 to 2016, Italy was identified as the main producer profile and exporter sub-profile, Spain and the

Netherlands as the main exporter profiles and producer sub-profiles and the rest of the European countries as unidentified profiles. At the second level of clustering applied, Germany and the United Kingdom were identified as importer sub-profiles, and France and Belgium as exporter sub-profiles. As a second selection criterion, COMTRADE's Resource Trade Earth [30] tool was used, which explores the dynamics of the rapid evolution of international trade in natural resources, the sustainability implications of such trade and the related interdependencies that arise between importing and exporting countries and regions. This tool allows the rapid identification of the main supplier (exporting) and consumer (importing) countries of tomatoes in the European market. Among the intra-community economies, the most important exporters are Spain, the Netherlands, France, Belgium, and Italy and as extra-community exporter, Morocco, in the European Union. On the other hand, Germany, the United Kingdom, France, Belgium, the Netherlands, Poland and Sweden stand out as importing markets. As a third criterion, the countries linked through some type of RTA subscribed in the World Trade Organization (WTO) were selected, according to the Integrated Trade Information Portal of RTA database WTO [31] with specific clauses for the active exchange of tomatoes. Morocco is incorporated as a non-EU supplier to the analysis of export competitiveness since Morocco has a RTA categorized as a Free Trade Agreement with the EU28 since 2000 and with an export quota to the European market above 4% in value in 2016.

### 3.3. Selection of time periods in the analysis due to the influence of the 2007/8 financial-economic crisis

The global economic-financial crisis of 2007/2008 negatively affected international trade in goods and services with a time lag of approximately one year. The negative effect on the international market was observed in 2009, with a positive "rebound" effect in 2010. With respect to temporal delimitation, the study of export competitiveness will focus on the period from 2005 until 2016, according to the latest consolidated data provided by COMTRADE [6]. The two-dimensional system of indicators will be analyzed in two consecutive time periods of equal temporal duration, pre-crisis (2005–2010) and post-crisis (2011–2016). In the periods of interest of this study on the European tomato market, it should be taken into account that in 2007 (pre-crisis period), two new countries were incorporated into the EU25, Bulgaria and Romania, forming the EU27, and finally in 2013 (post-crisis period) Croatia was annexed constituting the EU28. Finally, the CMS methodology was applied to data in two consecutive periods (2005–2010 and 2011–2016), in order to check the influence of the worldwide crisis on the change in the export volumes of tomatoes to the European market.

### 3.4. Constant market share

The CMS analysis is a methodology frequently used in the study of patterns of structural change in international trade. This analysis allows the relative contribution of competitiveness and structural factors of geographical and sectoral destination in the export performance of a country, or a group of countries, to be measured. It basically consists of differentiating the trade date of a given country (or group of countries) and comparing them with flows from the rest of the world [32]. The main idea behind this methodology is to show how the export market share of a reference country varies over a determined period if said country maintains the same share for all goods for all markets.

The CMS method was introduced by Tysznskin [33] and Richardson [34]. Subsequently, Ahmadi-Esfahani [35] adapted Jepma's version [17] to apply it specifically to the analysis of the export of agricultural products to specific markets, studying the case of variations in exports of Australian wheat to the Japanese market. The CMS method analyses the rate of

variation in export trade flows of a country over a certain period of time and breaks down said variation in two basic effects, one associated with demand forces and the second with supply variables [32]. Most competitive analyses of fruit and vegetable exports to the European Union market that use CMS methodology are undertaken by only from the supply perspective (analyzing the group of competing exporting countries). Thus, the work of Molina and Taiariol [27], who presented an export competitive analysis of citrus-producing countries (Argentina, Spain, Morocco, Uruguay and South Africa) that export or sell to the European target market, stands out. Initially, a breakdown was undertaken of the change in exports of the group of competitors to the European market (supply perspective), considering the European Union as a sole importing trade bloc. And subsequently, in a second stage, the competitiveness study was repeated, but from the perspective of differentiation of demand. Although it was only undertaken for one of the competing countries (Argentina) in the main client countries (Netherlands, Spain, the United Kingdom, France and Germany) that make up the European bloc, presenting a broader panorama of export competitiveness but not completed.

Starting out from the definition of a country's market share S = q/Q, $q$ being the export volume of a country with respect to a certain product and $Q$ the total volume of the same product traded worldwide and the result, $S$, as this country's worldwide market share. And differentiating the exports to the target market q with respect to time, its breakdown is obtained in two effects, the structural effect and the competitiveness effect. With the limitation that this equation is only valid in infinitely short periods of time, but if the breakdown is applied at discrete intervals [0.1], the breakdown of the exports to the target market $q$ incorporates a dynamic element in the analysis, the interaction effect (Equation 1 in Table 1). The interpretation of each effect at the first level of breakdown of the export volume was described by Leamer and Stern [36] and is shown in Table 1 along with the contributions of Valls [37]. In turn, each previous effect can be differentiated into two complementary effects (Equation 2 in Table 1).

Finally, The CMS methodology allows work in both physical units as well as monetary value, but it was decided to undertake the studies in terms of exchange port volumes, or sales of physical units, to avoid the use of economic deflators.

## 3.5. Normalization of competitiveness indexes

The design of multidimensional index systems requires the integration of a wide range of indicators with dissimilar characteristics. Methodological guides to implement index systems are scarce by the Economic by Economic Commission for Latin America and the Caribbean (CEPAL) but only provide general guidelines for their construction [39]. Despite the lack of instructive guides for the construction of multidimensional systems, an adaptation of the guide for the construction of composite indexes produced by the Organization for Economic Co-operation and Development (OECD) can be used. The "Handbook on constructing composite indicators" [40] by Joint Research Centre-European Commission recommended that normalization of indicators should be carried out to render the variables comparable. The normalization of indexes means adjusting values measured on different scales to a notionally common scale, without distorting differences in the ranges of values. As the first step, it is necessary to select suitable normalization procedure that respect both the theoretical framework and the data properties. Then, attention needs to be paid to extreme values and discuss the presence of outliers in the dataset as they may become unintended benchmarks. After that, to make scale adjustments and to transform highly skewed indicators, if necessary. In this case of study was selected a sigmoidal normalization (Eq 3) in order to rank between [+1, -1] the

**Table 1. Interpretation of different factors in the first and second level of exports decomposition of a region, country or commercial area makes in a target market.**

| | | | |
|---|---|---|---|
| First breakdown level | $\Delta q = Sj0\Delta Qj + Qj0\Delta Sj + \Delta Qj\Delta Sj$ (1) | Sj0ΔQj Structural Effect | Expected change in the volume of expected exports (ΔQ) supposing that the worldwide market share, and in the target market of the country under analysis, remain constant S, is the growth in exports exclusively driven by a growth in demand. |
| | | (+/-) signs indicate that the growth in demand for this product positively/negatively affects the growth in exports. In other words, if (+) is the growth in demand for the product, j positively affects the growth in exports. | |
| | | Qj0ΔSj Competitiveness Effect | Difference between real and expected exports and associated with changes in competitiveness, in other words, it is the growth in exports through displacement of competitors from the target market, interpreted as a relative improvement in the competitive profile. |
| | | (+/-) signs indicate that the country is gaining/ losing competitiveness. | |
| | | ΔQj ΔSj Interaction Effect | Measures the influence of interaction between changes in market share with changes in demand, in other words, the adaptation of market shares to the growth in imports of the markets supplied. |
| | | (−) sign indicates that exporters have lost market share in dynamic markets and gained market share in less dynamic markets. | |
| Second breakdown level | $\Delta q = ST0\Delta Qj + Sj0\Delta Qj - ST0\Delta Qj + Qj0\Delta ST$ $+ Qj0\Delta Sj - Qj0\Delta ST + QT1QT0 - 1Qj0\Delta Sj$ $+ \Delta Qj\Delta Sj - QT1QT0 - 1Qj0\Delta Sj$ (2) | ST0ΔQj Growth Effect | Expected change in export volumes (ΔQ) supposing that worldwide market share remains constant, is export growth attributed to demand for the product j. |
| | | (+) sign indicates an increase in demand for the product analysed on a worldwide level. | |
| | | Sj0ΔQj-ST0ΔQj Market Effect | Additional change expected in export volume (ΔQ) supposing that market share remains constant in the target market, is the growth in exports driven exclusively by a growth in demand for the product j in the target market. |
| | | (+) sign indicates that the exporting country tends to concentrate its exports to markets that are growing rapidly. | |
| | | Qj0 ΔST General Competitiveness Effect | Expected change in exports attributable to changes in competitiveness in general on a global level. |
| | | (+/-) indicate that the country is gaining / losing competitiveness at global level. | |
| | | Qj0 ΔSj-Qj0 ΔST Specific Competitiveness Effect | Expected change in exports attributable to changes in the specific competitiveness of the target market. |
| | | (+/-) indicate that the country is gaining / losing competitiveness at a specificat specific market. | |
| | | QT1QT0-1Qj0 ΔSj Pure interaction Effect | Measures the interaction between changes in the share of an exporter in the target market and changes in demand on a worldwide level. |
| | | ΔSjΔQj-QT1QT0-1Qj0 ΔSj Residual interaction Effect | Measures the interaction between changes in the share of an exporter in the target market and changes in the level of demand in the target market. |
| Where: Δq | | Breakdown of the analysed competitor country's export volume | |
| | Sj0 | Analysed competitor country's market share S in the target market j (at moment $t_0$) | |
| | ΔSj | Absolute change rate Δ of analysed competitor country's market share S in the target market j (between $t_1$-$t_0$) | |
| | ST0 | Analysed competitor country's market share S in the global market T (at moment $t_0$) | |
| | ΔST | Absolute change rate Δ of analysed competitor country's market share S in the global market T (between $t_1$-$t_0$) | |
| | Qj0 | Competing countries group's export volume Q of in the target market j (at moment $t_0$) | |
| | ΔQj | Absolute change rate Δ of the competing countries group's export volume Q in the target market j (between $t_1$-$t_0$) | |
| | QT0 | Competing countries group's export volume Q of in the global market T (at moment $t_0$) | |
| | ΔQT | Absolute change rate Δ of the competing countries group's export volume Q in the global market T (between $t_1$-$t_0$) | |
| | QT1QT0-1 | Relative change rate of the competing countries group's export volume Q in the global market T (between $t_1$-$t_0$) | |

Source: own elaboration with data from Contreras-Castillo [38] and Valls [37].

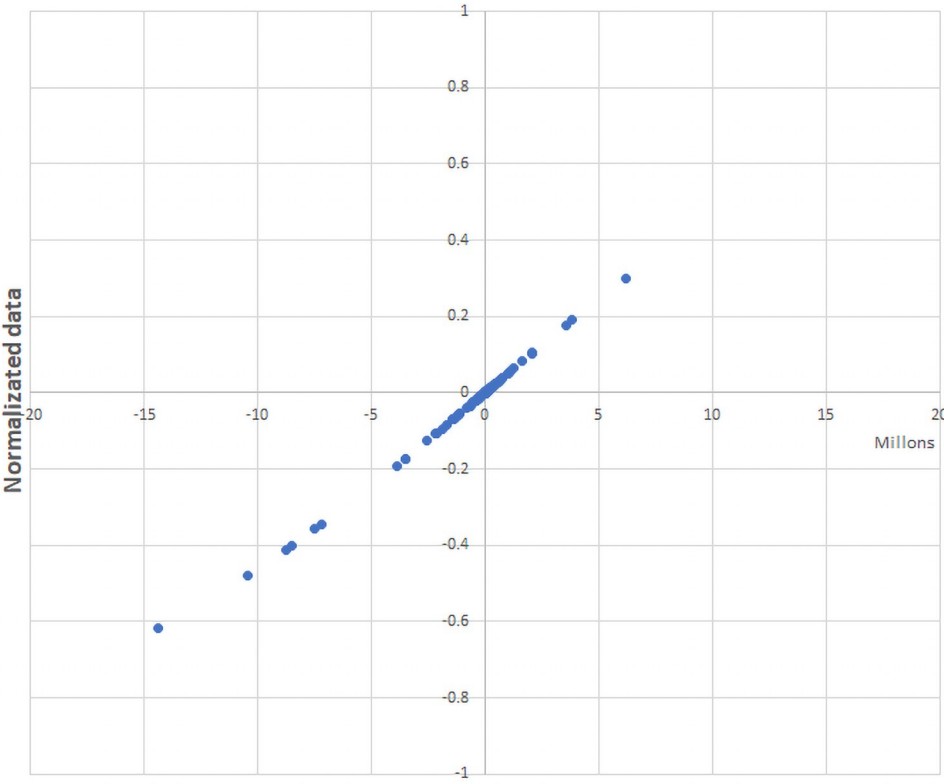

**Fig 11. Response curve of normalized data of general competitiveness.**

original competitiveness values obtained with the CMS methodology.

$$NCV = \frac{2}{\left(1 + e^{(-OCV*SF)}\right)} - 1 \qquad (3)$$

Where, NCV is the final Normalized Competitiveness Value, OCV is the Original Competitiveness Value and the SF is an appropriately selected scalar factor according to the order of the original data (SF = 0.0000001). Sigmoidal normalization was selected as a transformation function since the response curve of the normalized values of general competitiveness (Fig 11) and specific competitiveness (Fig 12) are located mainly in the linear zone, with no values located at the extremes that could generate saturated normalized values. Moreover, new outliers were not generated due to the normalization of the data.

Although the main purpose of the process of normalization in a multidimensional index system is to delimit the value rank, the process of normalization has helped to improve some normality characteristics of the data (Figs 13 and 14).

## 3.6. Export competitiveness maps

This research study has, as a methodological contribution, the presentation of the results by competitiveness maps. Competitiveness maps have the advantage of presenting the CMS competitive effect results as a 2D index system. Each competitiveness map was built taking the two variable components of the competitiveness effect, the general and specific competitiveness

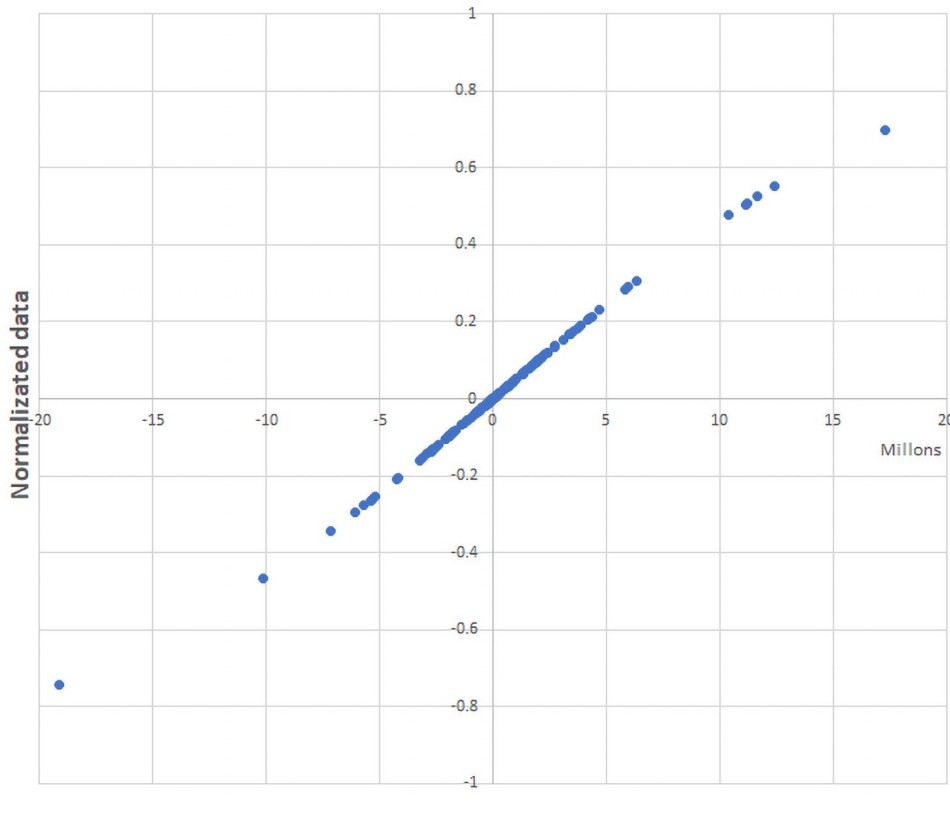

**Fig 12. Response curve of normalized data of specific competitiveness.**

effects, as the Cartesian axes. This type of representation has the advantage of showing the results together with the temporal directionality of the changes observed in the countries analysed, originating in its pre-crisis coordinates ($t_0$) and ending in its post-crisis coordinates ($t_1$). The ideal export competitiveness situation for an export economy is the location of its vectors in quadrant A, determined by both positive axes, along with a vectoral direction oriented towards the upper right corner.

This innovative competitiveness maps improve the results interpretation and offer complementary information on the evolution of each competitor in the target market during the pre-crisis and post-crisis period through the module, direction and meaning of its vector. As can be seen in Fig 15, the Cartesian axes of the 2D space, determine four quadrants coinciding with the different alternatives of general and specific competitiveness that can be adopted by the exporting-supplying countries in the main European tomato markets [41].

Group **A** is made up of highly competitive exporting countries, with positive values in both competitiveness indices, both in general and specific competitiveness, this quadrant is the ideal positioning of an exporting country in its target market. In contrast, Group **C** is made up of countries whose values in both indices are negative, being a non-competitive country. The other two groups, **B** and **D**, are made up of exporting economies that show only a positive value in one of the competitiveness effects of the second level (in general or in specific competitiveness). They are economies in the process of convergence, with the potential to become Group **A** economies if the directionality of the vector so indicates.

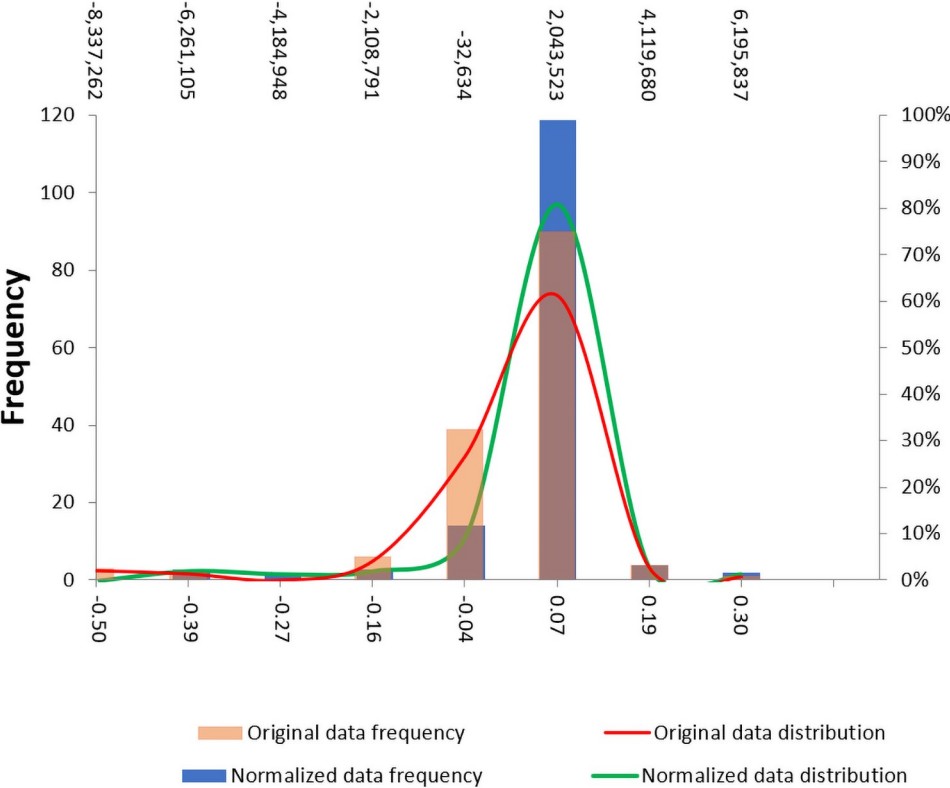

**Fig 13. Distribution of the general competitiveness indexes before and after the normalization.**

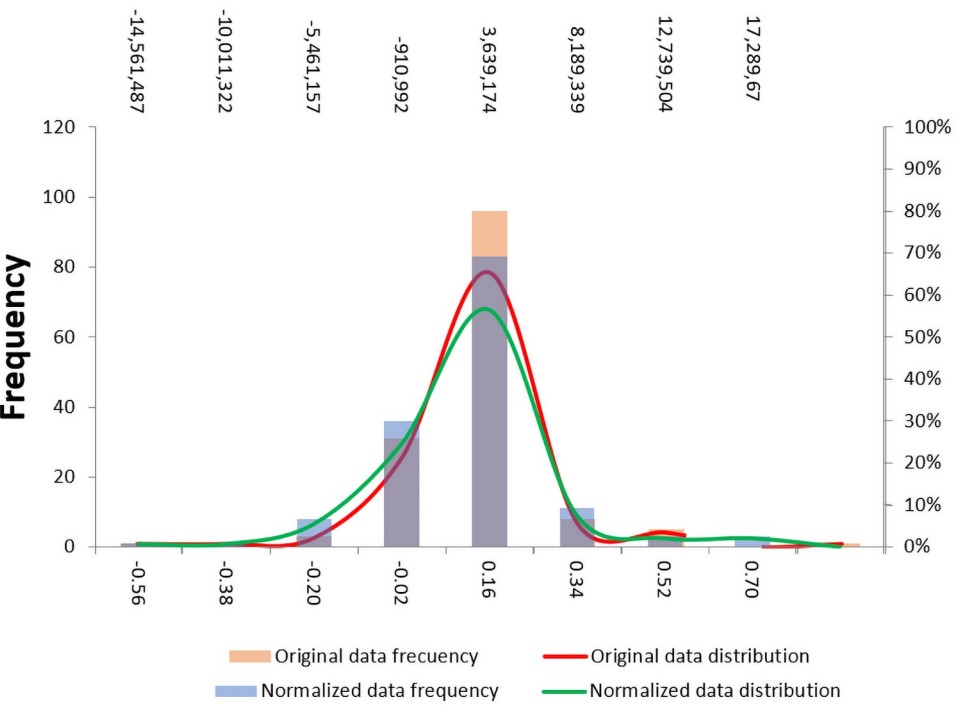

**Fig 14. Distribution of the specific competitiveness indexes before and after the normalization.**

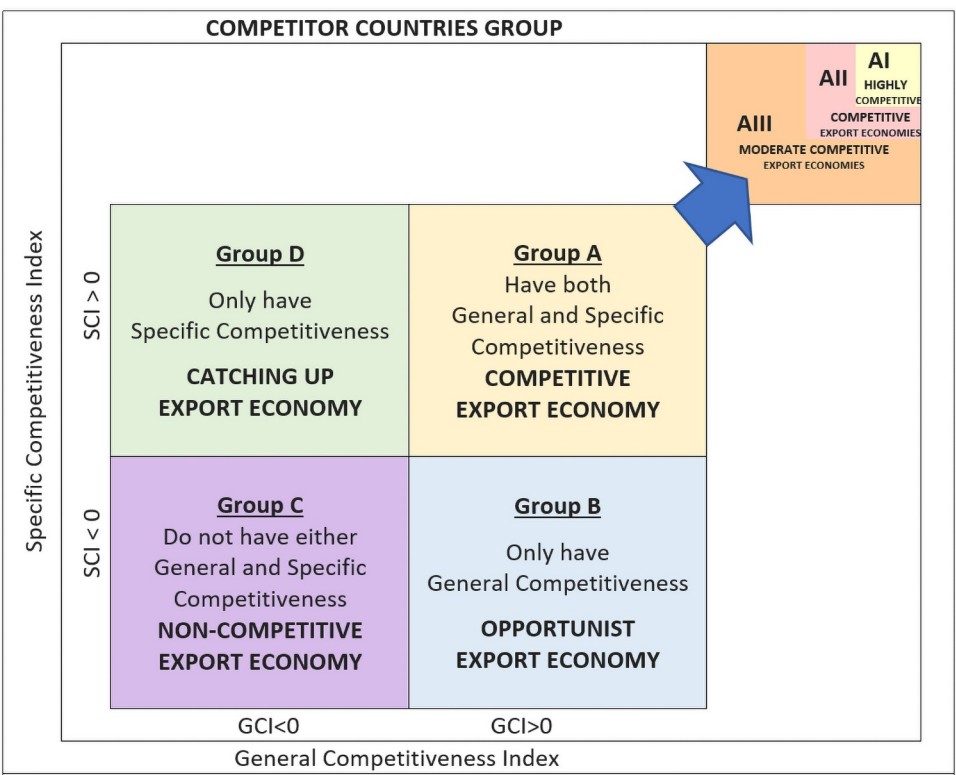

**Fig 15. Two-dimensional export competitiveness system for Constant Market Share.**

Group **A** includes the economies with the highest market share and/or competitiveness indexes in their regional market. The Group **AI** (highly competitive) of the leading exporting economies is formed by economies that are in first position in terms of market share of the target market. These leading economies will have to develop certain competitive strategies to maintain and even strengthen their positions [4]. The Group **AII** (competitive) is made up of challenging export economies, which are in second position in terms of market share. The challenging economy will have a double objective, trying to get closer to the leader/s to take their position, and keeping the followers away, so that none of them can take the challenging position. Group **AIII** (moderate competitive) is formed by economies with both positive competitiveness indexes, but their market shares represent less than 25% of the target market. These economies are called follower-competitive export economies.

Group **B** is made up of export economies that have positive general competitiveness indices but do not have specific competitiveness in the target market. These economies take advantage of their export structure towards the target market to place their non-competitive products in a timely manner.

Group **C** brings together export economies that are not competitive in the target market.

Finally, Group **D** is characterized by grouping exporting economies that are called emerging economies, because they have acceptable specific competitiveness rates but have not made the great leap to the target markets due to a lack of general competitiveness.

This research study has as a significant contribution, the temporal evolution of the system of indicators in a vectorial way, showing the values of the indicators together with the temporal directionality of the changes in the profiles observed in the supplier countries during both periods, pre-crisis and post-crisis. The intensity of the changes in competitiveness in the

economies analysed can be seen through the magnitude of the vector formed with the average values in each six-year period (pre- and post-crisis) of the indicators (general and specific competitiveness). This special representation, besides facilitating the interpretation of the CMS results, offers complementary information to the evolution of each competitor in the target market, through the direction and sense of the vector.

## 4. Results

From the initial studies by Ahmadi-Esfahani [35] to more recent studies [42], CMS methodology results are presented in numerical tables. These can be difficult to interpret for the purpose of drawing conclusions. By contrast, for the comparative analysis of periods, some authors use bar graphs, such as Rani et al. [43], Gonzalez et al. [44] and Capobianco et al. [3]. This graphic form enables an improvement in the interpretation of the breakdown of export volumes on the different levels that make up the components of competitiveness. This study presents a third level of analysis to achieve the objectives.

The objectives of this article are to obtain the competitive position between competitor countries in the main European markets and to determine whether there are post- crisis profile changes into the competitor exporting economies.

First, the bar graphs (S1 Appendix) allow the analysis of the temporal evolution of the components of competitiveness, namely specific and general competitiveness of the different competitors in the main European markets. Moreover, the export competitiveness maps show the temporal evolution of the competitiveness by vectorial format in the three main European markets, namely German, French and British markets (Figs 16, 17 and 18), whereby the most significant changes in export competitiveness for the competitor economies analysed between the pre-crisis and post-crisis periods are exhibited by the greater vectorial modules.

On the other hand, the Dutch, Belgian, Spanish, Italian, Swedish, and Polish and Belgian markets (S2 Appendix), with smaller vector modules, show a stable behaviour with insignificant changes in export competitiveness components among the competing countries and with a very centralized positioning in the axial origin. For this reason, only the analysis of each intra and extra-community competitor on the German, French, and British markets will be deepened.

In the German market (Fig 16), after the economic-financial crisis, only Spain, within the competitor group, presents a gain in specific competitiveness with positive value. Similarly, Belgium improves its competitiveness although without achieving positive value. The rest of the competitors lost competitiveness. Spain, like the Netherlands, are positioned as a catching-up economy in the German market map, with the difference that the Netherlands is in the transition zone towards a non-competitive economy. Belgium is positioned as a non-competitive economy and France as a moderate competitive economy in transition to a neutral profile (0,0).

In the French market (Fig 17), Spain as well as Belgium improved their competitiveness. Although Spain gains general competitiveness, but losses specific competitiveness, and it does not obtain positive values. By contrast, Belgium presents a less significant increase in general competitiveness but improves in specific competitiveness, managing to achieve positive competitiveness values. The French market map shows that Belgium and Spain have a non-competitive pre-crisis profile, where Spain maintains its non-competitive post-crisis and Belgium shows a post-crisis profile change to a moderate competitive export economy. It is worth noting that Morocco, with more than half of the French market, suffers from a post-crisis loss in both specific and general competitiveness. Thus, Morocco is in a transition process, leaving its

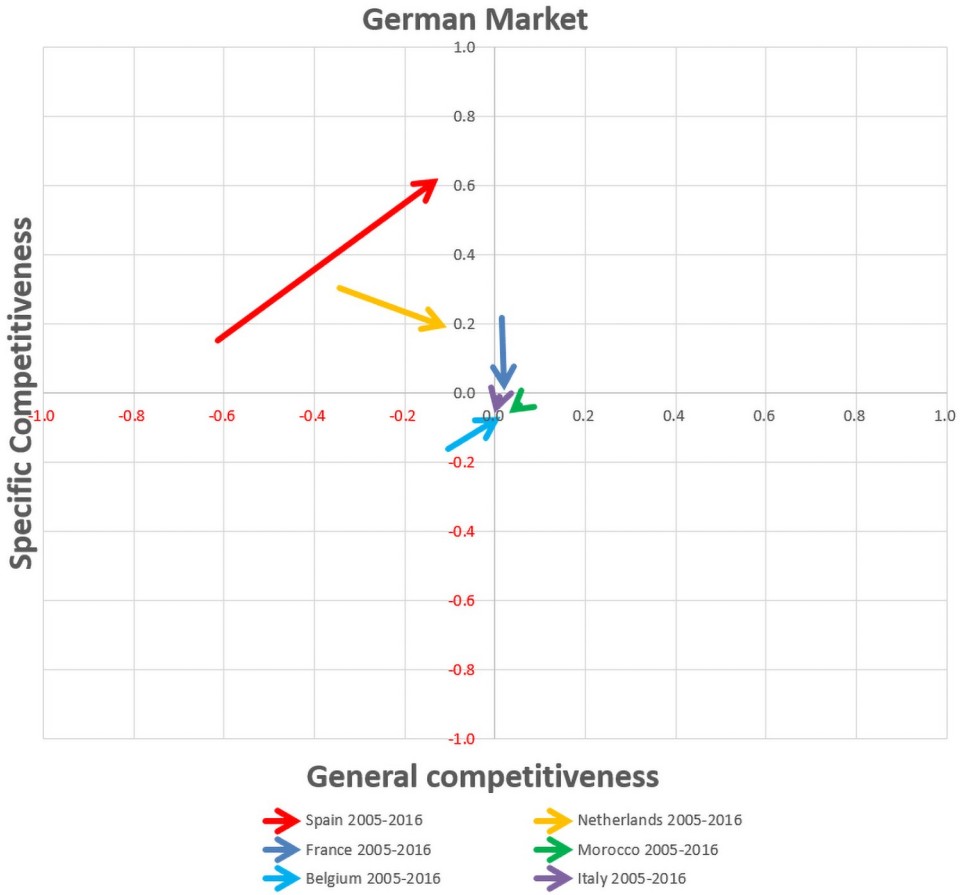

**Fig 16. Competitiveness map for competing countries group in the German market.**

moderate competitive economy in the French market. Finally, the Netherlands continues as a catching-up export economy.

With regards to the British market (Fig 18), only Spain presents an improvement in both general and specific competitiveness. Unlike Spain, France has increased its specific competitiveness but has decreased its general competitiveness. Finally, it is worth mentioning the loss in specific competitiveness in the Netherlands and Morocco, although the Netherlands gains in general competitiveness. Whereas Spain maintains its post-crisis profile as a catching-up economy, the rest of the competitors change their export profiles. Francia leaves its pre-crisis moderate competitive profile to a catching-up one. The Netherlands changes from a catching-up profile to a non-competitive profile. Both Morocco and Belgium are in transition zones from moderate competitive and catching-up respectively, to a neutral position.

In summary, Table 2 shows the post-crisis export profile changes seen in the group of competitors in the main European markets (German, French and British). Changes are observed in the Netherlands, France, Belgium and Morocco. In contrast, no changes are observed in Spain and Italy.

## 5. Discussion

The results offered by CMS methodology has allowed the display of results by export competitiveness maps along with traditional numerical tables and bar graphs in order to facilitate the

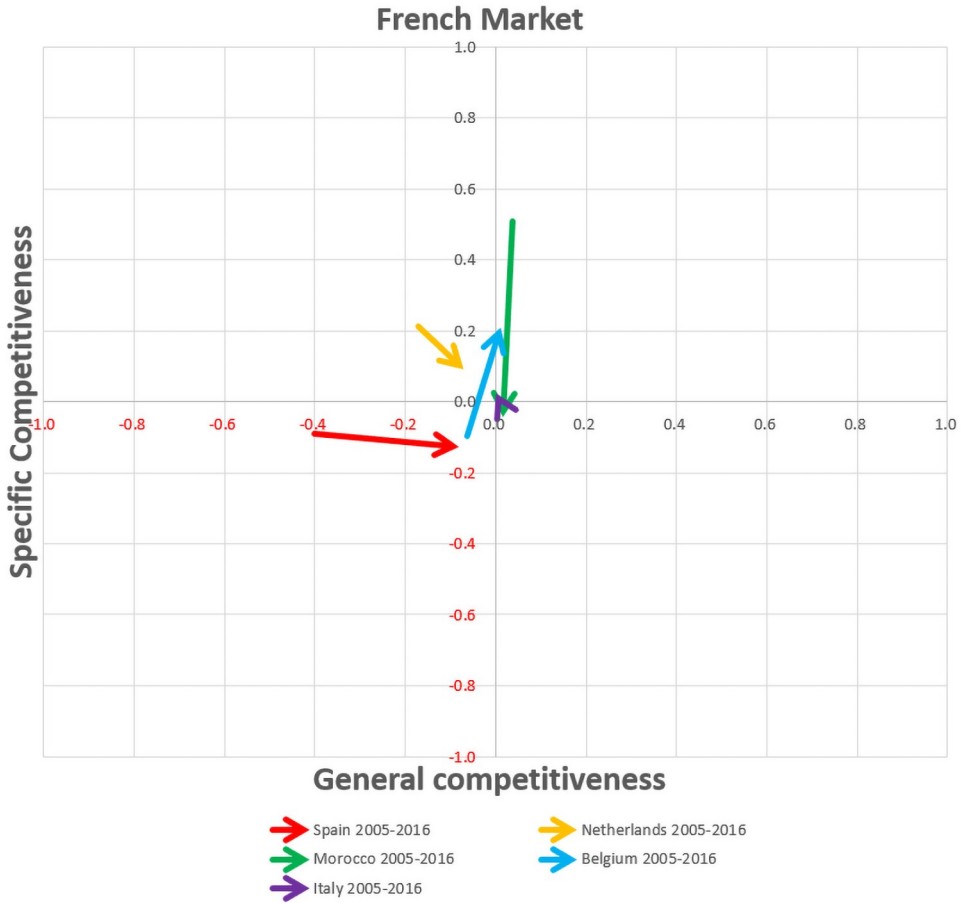

**Fig 17. Competitiveness map for competing countries group in the French market.**

interpretation of the components of competitiveness for competitors (general or specific) on each of its main European markets. Moreover, with the discriminated analysis of only three of the most important European markets, more than half of the intra-community market is covered (56%). The importance of the tomato as the vegetable with the greatest presence in international trade and European market as leading importer/exporter on a worldwide level of this vegetable, underline the relevance of undertaking a tomato export competitiveness study in the intra-community trade context, in order to analyze the possibilities of Spain to recover the lost leadership in 2009 against the Netherlands.

Germany is the main market of tomatoes in Europe as well as the main destination of intra-community Spanish sales [45], and currently remains so. Although the Netherlands continues being the leader in terms of export volume to the German market (Fig 7) and in pre-crisis period it was more competitive than Spain, in post-crisis period, the Netherlands has dramatically lost its specific competitiveness (Fig 16), being the Netherlands in the transition zone from a catching-up export economy to a non-competitive one (Table 2). In addition, after the crisis, Spain has increased both its general and specific competitiveness in the German market, continuing as a catching-up economy. This post-crisis situation may be beneficial for Spain in the forthcoming future if it continues increasing its competitiveness and, likely becoming a highly competitive economy in the German market. Especially if its main competitor continues to lose competitiveness along with Morocco and France (continue as opportunist export

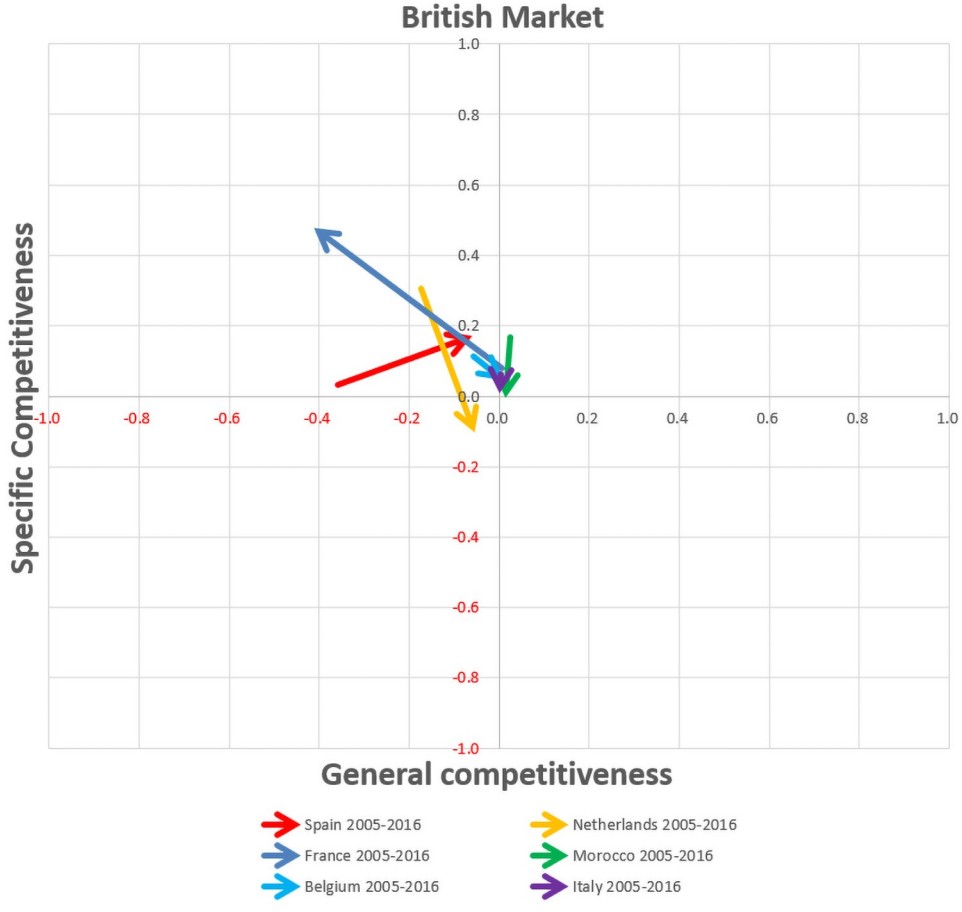

**Fig 18. Competitiveness map for competing countries group in the British market.**

**Table 2. Post-crisis export economy profile (2011–2016) for competitor economies in main European tomato markets.**

| | German market | | French market | | British market | |
|---|---|---|---|---|---|---|
| | Pre-crisis export economy profile | Post-crisis export economy profile | Pre-crisis export economy profile | Post-crisis export economy profile | Pre-crisis export economy profile | Post-crisis export economy profile |
| **Netherlands** | Transition zone (D→C) | | Catching-up (D) | | Change(D→C) | |
| | Catching-up | Non-competitive | | | Catching-up | Non-competitive |
| **Spain** | Catching-up (D) | | Non-competitive (C) | | Catching-up (D) | |
| **Morocco** | Opportunist (B) | | Transition zone (AIII→B) | | Transition zone (AIII→B) | |
| | | | moderate Competitive | Opportunist | moderate Competitive | Opportunist |
| **France** | Transition zone (AIII→B) | | | | Change (AIII→D) | |
| | moderate Competitive | Opportunist | | | moderate Competitive | Catching-up |
| **Belgium** | Transition zone (C→B) | | Jump (C→AIII) | | Transition zone (D→AIII) | |
| | Non-competitive | Opportunist | Non-competitive | moderate Competitive | Catching-up | moderate Competitive |
| **Italy** | Central axial (0,0) | | Central axial (0,0) | | Central axial (0,0) | |

Source: Own elaboration with data from COMTRADE [6]

economies). Nevertheless, Belgium's growth in general and specific competitiveness must be taken into account. Although, Belgian share in German market does not reach 10%, its commercial trend is oriented towards the region of competitive export economy. Nowadays, Belgium does not stand out for its volume of tomato production, although it has the highest productivity rates in Europe (Fig 8), the production of greenhouse vegetables has increased to 375 mt. The largest share of greenhouse vegetables (70%) concerns tomatoes and the Belgian tomato production has gradually increased to 260 mt [46].

In the French market, second in importance in the European market, Morocco is the leading supplier (Fig 7) and a competitive export economy (Fig 17) during pre-crisis period. However, since the economic-financial crisis, Morocco presents a great loss in its specific competitiveness, leaving the leadership in competitiveness and becoming an opportunistic export economy (Table 2). In the French market, Morocco is losing specific competitiveness and Spain has not managed to increase its specific competitiveness to take this opportunity. According to FEPEX [47], community imports of fresh Moroccan vegetables has undergone strong growth in recent years, driven by Morocco not applying the requirements demanded of community producers in multiple spheres, from environmental to social. The Spanish Administration, along with the French, are the two making the greatest efforts to achieve much more demanding regulations for products entering from third-party countries and applying the principle of reciprocity. Spain continues to be the leading supplier of fruit and vegetables to France and has earned an image of quality [48], although Morocco continues to gain market share thanks to its lower prices. In such a way, other topic that deserves attention in the context of international trade is the role of the private food standards and certifications, which are becoming increasingly important. In the European context, Global GAP certification is deemed as one of the most important and preferable for retailers [49, 50]. The compliance with Global GAP standards involves costs that can be barriers to countries with strategy via costs [51]. In the European context, Morocco is the competitor with strategy based on costs. In this vein, it should be interesting to analyse the percentage of extort production of tomatoes that is certified and required by French market since France is the main market for Morocco.

In the British market, the third largest in the European market which has recently left the intra-Community market, the Netherlands had the largest market share (Fig 7). In the British market post-crisis, the Netherlands has drastically reduced their specific competitiveness (Fig 18). In addition, Morocco and France were both competitive export economies in pre-crisis, but have lost specific and general competitiveness respectively, turning Morocco into an opportunistic export economy and France into a catching-up export economy. It is worth mentioning the widespread acceptance among British retailers of voluntary quality certifications. In this regard, Global GAP certification is regarded as the first step towards a European harmonised standard for integrated production, rooted in acts such as United Kingdom food safety of 1990 whereby emerged a shift in responsibility [52].

Deepening in the underlying factors of competitiveness so as to foresee the future competitiveness, several factors should be examined. According to Contreras-Castillo [38], the competitiveness of a product in the international market depends on its comparative advantages associated with favourable natural factors and lower relative production costs. The author indicates that there are other factors that may influence changes in competitiveness, such as an improvement in product quality, the rate of exchange, the degree of product differentiation, the efficiency of trading and financing services, transport costs and discriminatory trade policies.

In this sense, trade policies basically consider two types of competitiveness strategies, via costs or product differentiation [4]. The strategy via costs allows to be competitive by selling in bulk and offering the products at low prices in the market. From the export price data

provided in the second section, Spain adopted the cost strategy in the pre-crisis period, but simultaneously was losing competitiveness against the Netherlands (Figs 16, 17 and 18; S2 Appendix) and finally lost the export leadership in 2009 against the Netherlands. After 2012, when Morocco started to export by a new Free Trade Agreement to the European Union, Spain had to change its strategy, focusing on product differentiation, since it cannot compete with Morocco regarding costs. In this respect, Morocco has a more competitive production cost structure with lower labour costs (1.12 euros/hour) than Spain (8.37 euros/hour) [53]. It is worth noting that the Netherlands stands out in the post-crisis period as the leading country, with its strategy of purchasing in bulk, re-exporting and splitting up the product presentation, which has allowed the increase of exports and value-added products, even without having its own production of tomatoes for internal supply and maintenance of its client markets (Fig 8).

In addition, the differentiation strategy may be based on quality requirements. In the last years, the agricultural food sector in Europe has shown an increase in the demand for quality safe food to ensure producers to comply with quality and safety requirements [54]. In this regard, mandatory and voluntary quality assurance standards are deemed to pressure producers to meet food safety management systems [52]. Accordingly, Global Gap highlights in Europe with a widespread acceptance. Moreover, in European Union market, it is thought to be a *de facto* requirement for most major European retailers. In this regard, intra-community competitors have gone steps further to ensure production meets safety and quality requirements in comparison to Morocco [54].

The Food Foundation experts [55] examined the potential impact of Brexit scenarios on the price of fruit and vegetables. They exposed that the inflation resulting from unfavourable exchange rates and higher import prices, and the rising cost of seasonal labour would lower the standard of living for United Kingdom residents. They looked at the self-sufficiency levels of fruit and vegetables and grouped them according to their trade characteristics and therefore the impact which new trading rules could have on them. Tomatoes make up the "The Brexit Boosters" group, products which are grown in the United Kingdom but also rely on European imports, whose production could increase if the United Kingdom became more competitive and/or if tariffs were introduced for European Union's imports. Currently, the United Kingdom has a medium level of self-sufficiency (19,64%) in fresh tomatoes and a value of m£744.30 in the United Kingdom market. The largest British producer of tomatoes has announced its expansion, with investments in greenhouses on the Isle of Wight, where it will also reserve more space for organic production [56]. On the other hand, it should also be considered that the United Kingdom is already vulnerable because heat waves and droughts caused by global warming have reduced local food production [57].

The increase in Spanish competitiveness in the German, French, and British markets highlights a rise in general over specific competitiveness (Figs 16, 17 and 18). The rise in competitiveness indicates a growth in Spanish exports through displacing competitors from the target market. In the German market, Spain could displace the Netherlands. The "cherry" category is the most highly tomato variety valued by the German market [58], which is why Spain should extend its presence in this market and present higher value-added products to match Dutch cherry tomato prices. The British market is one that sets trends in the consumption of fresh fruit and vegetables [59] and presents the highest concentration of single-person households in Europe. Spain should take advantage of the double opportunity of Netherland's loss of specific competitiveness and France's loss of general competitiveness. In this market, Spain should compete with colour cherry tomatoes for snacking on outside the home and with small pack sizes by unit in the rest of the varietal categories. Finally, in the French market, although its main competitor is Morocco, Spain could take advantage of the specific competitiveness loss of Morocco and support differentiation strategy due to the fact that Spain is not competitive in

prices in comparison to Morocco. In this regard, Spain can benefit from the growing interest in products sourced from ecological agriculture. France is the third largest organic market [60]. Although this trend is not exclusive to the French market, according to the report by Van Rijswick [2], organic foods are gaining market share all over the world. The organic tomato market can generate a substantial increase in Spain's general competitiveness, as it has the largest organic agricultural areas in Europe, almost 2 million hectares [60].

Tomatoes, oranges, and apples are three examples of the UK's dependence on the European Union for fruits and vegetables [61] and have viable non-EU sources. While United Kingdom tomato imports are currently dominated by the EU, Morocco has exported increasing amounts of the commodity to the country over the past ten years, according to Her Majesty's Revenue and Customs [62]. In a broader context, pursuing these alternative sources will allow the United Kingdom to begin to develop bilateral trade. Trade agreements with large, developed agricultural countries like the US and China would take a long process of negotiations that the United Kingdom cannot afford. Other possibility is to join NAFTA, opening an important transatlantic market to Mexican and American tomatoes. Although the United Kingdom is likely to impose entry barriers to genetically modified fruit and vegetables. Another point to consider for European competitors on the British market is the fact that its specific competitiveness will decrease. Not only because of the increase in prices due to the abolition of Britain's tariff-free trade status with the other European Union's members [7], but also because of the elimination of subsidies that compensate for the extra cost of transport to the United Kingdom. This type of aid is only applicable to the transit of goods between European Union member countries and not to exports to third countries.

Spain will also achieve the British market, but this trend will depend on the conditions of bilateral trade between the United Kingdom and the European Union (EU27) post-Brexit. Finally, the Spanish competitiveness in the German market will reach the group A of competitive leadership in the long term. Adding together all the contributions of Spanish increased competitiveness in the main European markets, it is possible to foresee a recovery of the leadership in volume exported of tomatoes in the European market.

## 6. Conclusions

In the pre-crisis period, Spain was the leader in tomato sales to the European Union, despite having the lowest general and specific competitiveness indicators of the entire group of competitors. At the same time, the Netherlands had better values in general and specific competitiveness and, in 2009 became the leader in tomato exports, dethroning Spain. Applying the CMS methodology for the analysis of the components of general and specific competitiveness, this study highlights the possibility of Spain to regain leadership, through the profiles of post-crisis export offered by the competitiveness maps.

In the post-crisis period, the Netherlands has lost competitiveness in all European markets. In the German market, the Netherlands is in the transition zone between catching-up to non-competitive export economy and presents a change of profile from catching-up to non-competitive in the British market. On the other hand, while Spain is a catching-up economy in the German and British market in post-crisis period, in the French market stands as a non-competitive economy. Nevertheless, the loss of competitiveness presented by Morocco may turn out to be an opportunity for Spain by a product differentiation strategy since Spain. In the group of competitors, only Belgium together with Spain are the candidates for becoming fully competitive countries.

As a future line of this study, the British market should be analysed post-Brexit due to its importance as a client market in Europe and Spain in order to analyse the resulting supply and

demand restructuration of the European tomato market. What is more, these "export competitiveness maps" obtained through a two-dimensional system of indicators could be extended to a three-dimensional system, with the incorporation of a CMS third indicator about the dynamism of customs markets.

## Supporting information

**S1 Appendix.**
(DOCX)

**S2 Appendix.**
(DOCX)

## Author Contributions

**Conceptualization:** María de las Mercedes Capobianco-Uriarte, Jaime De Pablo-Valenciano, María del Pilar Casado-Belmonte.

**Data curation:** María de las Mercedes Capobianco-Uriarte.

**Formal analysis:** María de las Mercedes Capobianco-Uriarte, Juan Aparicio.

**Investigation:** María de las Mercedes Capobianco-Uriarte, Juan Aparicio.

**Methodology:** María de las Mercedes Capobianco-Uriarte, Juan Aparicio, Jaime De Pablo-Valenciano.

**Project administration:** María de las Mercedes Capobianco-Uriarte.

**Resources:** María de las Mercedes Capobianco-Uriarte, Juan Aparicio.

**Software:** María de las Mercedes Capobianco-Uriarte.

**Supervision:** María de las Mercedes Capobianco-Uriarte, Juan Aparicio, Jaime De Pablo-Valenciano, María del Pilar Casado-Belmonte.

**Validation:** María de las Mercedes Capobianco-Uriarte, Juan Aparicio, Jaime De Pablo-Valenciano, María del Pilar Casado-Belmonte.

**Visualization:** María de las Mercedes Capobianco-Uriarte, Juan Aparicio, Jaime De Pablo-Valenciano, María del Pilar Casado-Belmonte.

**Writing – original draft:** María de las Mercedes Capobianco-Uriarte, María del Pilar Casado-Belmonte.

**Writing – review & editing:** María de las Mercedes Capobianco-Uriarte, María del Pilar Casado-Belmonte.

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
