## [Decision Letter · Decision Letter 0]

8 Jul 2020

PONE-D-20-05952

THE LIKELIHOOD OF SPAIN RECOVERING ITS PRE-CRISIS LEADERSHIP IN THE EUROPEAN TOMATO MARKET.

AN APPOACH USING EXPORT COMPETITIVENESS TRENDING MAPs

PLOS ONE

Dear Dr. Casado-Belmonte,

Thank you for submitting your manuscript to PLOS ONE. After careful consideration, we feel that it has merit but does not fully meet PLOS ONE’s publication criteria as it currently stands. Therefore, we invite you to submit a revised version of the manuscript that addresses the points raised during the review process.

I must stress that while I am offering you an opportunity to revise and resubmit this paper, I am not able to provide a guarantee that I will be able to publish it should you decide to take up our offer and resubmit a revision that deals with the comments in the attached referee reports.

More precisely we ask you to take into account  the detailed comments of the referees 3 and 4 (and to a lesser extent comments from referees 1 and 2).

A rebuttal letter that responds to each point raised by the reviewers . You should upload this letter as a separate file labeled 'Response to Reviewers'.A marked-up copy of your manuscript that highlights changes made to the original version. You should upload this as a separate file labeled 'Revised Manuscript with Track Changes'.An unmarked version of your revised paper without tracked changes. You should upload this as a separate file labeled 'Manuscript'.

We look forward to receiving your revised manuscript.

Kind regards,

Damien Rousselière, PhD

Academic Editor

PLOS ONE

Journal Requirements:

2. Please remove your figures from within your manuscript file, leaving only the individual TIFF/EPS image files, uploaded separately.  These will be automatically included in the reviewers’ PDF.

3. We note you have included a table to which you do not refer in the text of your manuscript. Please ensure that you refer to Tables 3,4,5 & 6 in your text; if accepted, production will need this reference to link the reader to the Table.

Reviewers' comments:

Reviewer's Responses to Questions

**Comments to the Author**

1. Is the manuscript technically sound, and do the data support the conclusions?

Reviewer #1: Partly

Reviewer #2: Partly

Reviewer #3: Partly

Reviewer #4: Partly

2. Has the statistical analysis been performed appropriately and rigorously? 

Reviewer #1: I Don't Know

Reviewer #2: N/A

Reviewer #3: I Don't Know

Reviewer #4: Yes

3. Have the authors made all data underlying the findings in their manuscript fully available?

Reviewer #1: Yes

Reviewer #2: Yes

Reviewer #3: No

Reviewer #4: Yes

4. Is the manuscript presented in an intelligible fashion and written in standard English?

Reviewer #1: No

Reviewer #2: No

Reviewer #3: No

Reviewer #4: Yes

5. Review Comments to the Author

Reviewer #1: Attached. In summary, I do not recommend the publication of this paper. The effort made by the authors in compiling graphs and tables is outstanding, but its academic value-added and the innovative character of its contribution to the literature is not clear. Economically, the authors present a comprehensive descriptive and accounting exercise, but they do not elaborate on the reasons behind the changes observed in specific and general competitiveness. Simply applying a methodology that already exists and presenting nice graphs is not enough as academic value-added. More modelling of the trends observed would be needed to increase the paper’s scientific value. Finally, the English language used desperately needs editing by a native speaker, and further attention should be devoted to detail (e.g. even the title of the paper presents mistakes: 'APPOACH' and 'MAPs').

Reviewer #2: The manuscript is not presented in an intelligible fashion and written in standard english.

the introduction is not clear in terms of research questions and objectives. readers do not understand which are the main scopes of the articles and there is not a clear research question expressed.

The focus county is Spain but why? in other words, why Spain is interesting in the international scenario of tomatoes market? this aspect is not clear.

Reviewer #3: The methodology and the graphical solutions hypothesised are interesting, as well as the descriptive picture derived from indicators. However:

1) the use of only quantities traded does not provide enough evidence for the Result and Conclusions sections. CMS is a descriptive method. Its enlargement to demand factors weakens the approach if there is no room for quality related souces of competitive advantage to be ascertained. The role of price, at least, in the different importing countries should be explicitly dealt with. Also other relevant dimensions, such al logistic, SPS standards etc. should be considered and documented.

An alternative could be to go back to supply only, integrate the numbers with information on the different players and have a simplified and sounder picture and discussion.

2) The statistical analysis is probably performed appropriately and rigorously. However, both limitations in the implementation of Plos Data policy (see point 3 below) and a seriously lacking presentation of tables and graphs (see point 4 below) does not allow to be sure of that.

3) Original Comtrade data used for the indicators are not provided in the annexes to the manuscript.

4) The manuscript is written in standard English. However:

- unit measures in some tables are missing or wrong. Title and labels of tables are often missing, for both table 1 in the text and the other tables.

- the definition of variables (p. 10-11) is unclear, the disctinction between world flows and target country's flows is mainly a matter of intuition. This also makes unclear the meaning of the adopted breakdowns.

- Label and colors in the map are confusing. To be checked and revised making them more consistent throughout the maps

- pages from 12 to beginning 14 are repetitious and can be synthesized

- some editing work throughout the text is recommended

Reviewer #4: General comments

This article deals with the analysis of tomato export competitiveness. The article focuses on Spain position according to its main competitors. The methodology used is the Constant Market Share (CMS) approach. The results are presented in a new graphical way using trending maps.

The article highlights important results and develops an interesting discussion. Nevertheless I would suggest some improvements before publication. The improvements I suggest are threefold:

-First, I am not sure the methodology of the CMS is properly presented. The current presentation in table 1 is quite difficult to follow. The variables used in the equations are poorly explained and the indices (j or j0, T0….) are not explained. At the end, I was lost! And this is very important that the reader to fully understand the methodology developed.

-Second, I find the mapping of the results very interesting; and I agree, this is much easier for the reader to understand the results. The analysis of the results gives an overview of the evolution of the export competitiveness in the tomato market for different countries. But I am not convinced that the analysis of the maps can give any idea of what will happen in the next period. The author(s) mention that, based on the results, we can ‘foresee” some evolutions. I am not convinced. To do so, I would have expected an analysis of the underlying factors explaining the trends at play. For instance the author(s) discuss about the role of cherry tomato. I appreciate this discussion. But I would expect a deeper discussion on the determinants of the potential evolution of the productions in the different competitor countries. Such a discussion would have been helpful, especially in the case of Spain (as it is the country under interest). The same remarks holds for the different determinants of consumption of “cherry tomato” in the different target countries.

-Third, as the author(s) work on the tomato product, I was expecting a discussion on the role of private standards and especially GLOBAL GAP. Is there a link between the trend observed and the certification with GLOBAL GAP? At the end, I was very surprised that this important determinant of imports (linked to the requirement of retailers) was not accounted for in the analysis. I think this is a very important element to include in the article.

Detailed comments

-The equations in table 1, especially equation 3 should be explained in details.

-Normalization: some details should be given on the normalization procedure. What is the distribution of the indexes before and after the normalization?

-P.12, l. 296, it is mentioned “future commercial trends”. I’m not convinced about this. Please develop!

-P.13 description of group AI, AII, AIII. These groups are not mentioned on figure 12. A clear description should be given (with number or location on the graphic).

-Results: sometimes confusing. As Spain is the country of interest, may be the discussion should be centered on it. I wonder whether this article could not have a wider scope than mainly Spain. Especially if you succeed in deepening the analysis on the determinants of the highlighted trends.

---

## [Decision Letter · Decision Letter 1]

15 Jan 2021

PONE-D-20-05952R1

THE EUROPEAN TOMATO MARKET.

AN APPROACH by EXPORT COMPETITIVENESS TRENDING MAPS

PLOS ONE

Dear Dr. Casado-Belmonte,

Thank you for submitting your updated manuscript to PLOS ONE. After careful consideration, we feel that it has merit but does not fully meet PLOS ONE’s publication criteria as it currently stands.  Therefore, we invite you to submit a revised version of the manuscript that addresses the points raised during the review process.

- please adress the concerns of reviewer 4 about the "transparency" of your forecastings, as transparency and reproductibility are among the major criteria for publication in PLOS ONE. Explain better the way the predictions are estimated and the limits of your methodology. 

-  focus the rewriting on the most interesting cases and provide an additional online appendix for the other materials (see comments 3 of referee 4)

- follow the advices of reviewer 4 about the abstract and the introduction

We look forward to receiving your revised manuscript.

Kind regards,

Damien Rousselière, PhD

Academic Editor

PLOS ONE

Reviewers' comments:

Reviewer's Responses to Questions

**Comments to the Author**

1. If the authors have adequately addressed your comments raised in a previous round of review and you feel that this manuscript is now acceptable for publication, you may indicate that here to bypass the “Comments to the Author” section, enter your conflict of interest statement in the “Confidential to Editor” section, and submit your "Accept" recommendation.

Reviewer #1: All comments have been addressed

Reviewer #3: All comments have been addressed

Reviewer #4: All comments have been addressed

2. Is the manuscript technically sound, and do the data support the conclusions?

Reviewer #1: Yes

Reviewer #3: Yes

Reviewer #4: Yes

3. Has the statistical analysis been performed appropriately and rigorously? 

Reviewer #1: Yes

Reviewer #3: Yes

Reviewer #4: Yes

4. Have the authors made all data underlying the findings in their manuscript fully available?

Reviewer #1: Yes

Reviewer #3: Yes

Reviewer #4: Yes

5. Is the manuscript presented in an intelligible fashion and written in standard English?

Reviewer #1: Yes

Reviewer #3: Yes

Reviewer #4: Yes

6. Review Comments to the Author

Reviewer #1: This version of the paper has been reviewed substantially. It incorporates all the comments and suggestions made to the original version. I particularly appreciate the clarity in the individualized response to each one of them.

This version substantially improves the clarity and consistency of the general message.

The authors presented in the first version a comprehensive, descriptive, and accounting exercise. In addition, this revised version of the paper introduces a thorough discussion of the reasons behind the changes observed in specific and general competitiveness. This critically adds value to the paper.

Consequently, I recommend the publication of the paper.

Reviewer #3: (No Response)

Reviewer #4: 1/The new version of the article accounts for most of my comments on the first version.

Table 1 has been improved in a relevant way. A section on GlobalGap has been added. I note that it was mentioned following the French market analysis. It is important to mention also that the UK, as an important fruit and vegetable importer, was the first to impose GlobalGap, mainly through its retailers, which are the main sellers of fruit and vegetable to the British consumers.

2/Regarding, the methodology and analysis, a new development appears to deal with the “foreseeing issue” of the paper. Initially the author(s) wrote they could do prediction in the future of competitiveness on the tomato market. I did not agree with this point. To account for this issue, in the new version, the authors mentioned that their methodology shows the evolution between periods, but also the export future trend, obtained by linear regression applied a linear regression model in each economy. This is mentioned in the authors’ answer to my comments. I need more explanations!

For me, this means that the authors only consider that the previous evolution will be the same as the future one. In other words, this evolution will be the relevant one if all factors explaining the previous evolution will stay the same in the future as before. As mentioned by the authors, the Brexit will have an impact in the coming years. Thus, I cannot agree with this way to proceed.

In my understanding, the exact determinants of the previous evolutions are not clearly measured, it is difficult to “foresee” something different in trends. I think this is not interesting to add these aspects. I would suggest to the authors to try to explain better the main determinants of the previous evolution.

3/The current version addresses a wider scope than the previous version. I think it is a relevant point. However, this leads to a too long article, with too many Figures (30 figures!!!). I would suggest to provide an online appendix for instance, with all figures available. The figures proposed in the article should focus on one or two detailed cases, the most interesting cases. The discussion should be based on all the figures available in the appendix. This would lead to a more straightforward reading. The current version gives lots of information, may be too much and it loses the reader. May be some key component of competitiveness should be the main component of the paper. For instance the discussion on cost competitiveness and the illustration with the relative position of Spain and Morocco is very interesting. It should be deepen and only the corresponding graphs should be included in the main texte of the article. The same holds for the discussion on private standards.

4/I suggest below some suggestions to improve this article. I think the article would gain a lot from a new editing with the objective of being more straightforward. Frist, the abstract is too long. I suggest to focus on the contribution of the article, mains the CMS analysis and the graphical representation of the results. The introduction has also to be reorganised in order to gain in efficiency. In the current version, the objectives of the article are presented on the 6th page. It is too late. From the first two pages, the reader expects an analysis on the impact of regional trade agreements on fruit and vegetable trade. It sounds more an “introductory chapter” than an introduction of a scientific paper. The reader needs to know on the first page what are the question, the contributions, and the literature strand in which the article is included. Another example: the data on prices given in the introduction are more relevant in the discussion of the results. It’s not usual to have some references to the introduction to explain some results at the end. I would suggest to be more straightforward in the introduction, and to use some materials later on.

At the end of my comments, I still appreciate the 2D mapping. I think it is interesting for the reader to have some graphical representations. This is a good way to have a clear overview of the competitiveness in a sector, with lots of countries concerned. Anyway, I think the article should gain in structuration and deepen the quantitative analysis of underlying factors, especially to foresee the future competitiveness. The discussions on prices, quality and private standards (as GLOBAL GAP) are very interesting and should be the main part of the discussion.

7. PLOS authors have the option to publish the peer review history of their article (what does this mean?). If published, this will include your full peer review and any attached files.

Reviewer #1: **Yes: **Enrique Martinez-Galan

Reviewer #3: No

Reviewer #4: No

---

## [Decision Letter · Decision Letter 2]

16 Apr 2021

THE EUROPEAN TOMATO MARKET.

AN APPROACH by EXPORT COMPETITIVENESS MAPS

PONE-D-20-05952R2

Dear Dr. Casado-Belmonte,

We’re pleased to inform you that your manuscript has been judged scientifically suitable for publication and will be formally accepted for publication once it meets all outstanding technical requirements.

Kind regards,

Damien Rousselière, PhD

Academic Editor

PLOS ONE

Additional Editor Comments (optional):

Reviewers' comments:

Reviewer's Responses to Questions

**Comments to the Author**

1. If the authors have adequately addressed your comments raised in a previous round of review and you feel that this manuscript is now acceptable for publication, you may indicate that here to bypass the “Comments to the Author” section, enter your conflict of interest statement in the “Confidential to Editor” section, and submit your "Accept" recommendation.

Reviewer #4: All comments have been addressed

2. Is the manuscript technically sound, and do the data support the conclusions?

Reviewer #4: Yes

3. Has the statistical analysis been performed appropriately and rigorously? 

Reviewer #4: Yes

4. Have the authors made all data underlying the findings in their manuscript fully available?

Reviewer #4: Yes

5. Is the manuscript presented in an intelligible fashion and written in standard English?

Reviewer #4: Yes

6. Review Comments to the Author

Reviewer #4: I appreciate this new version of the article.

It incorporates all the comments and suggestions made to the previous version, and all the answers provided to my questions are convincing.

I recommand the publication of the paper.

7. PLOS authors have the option to publish the peer review history of their article (what does this mean?). If published, this will include your full peer review and any attached files.

Reviewer #4: No

---

## [Editor Report · Acceptance letter]

22 Apr 2021

PONE-D-20-05952R2 

The European tomato market. An approach by export competitiveness maps 

Dear Dr. Casado-Belmonte:

I'm pleased to inform you that your manuscript has been deemed suitable for publication in PLOS ONE. Congratulations! Your manuscript is now with our production department. 

Kind regards, 

on behalf of

Dr. Damien Rousselière 

Academic Editor

PLOS ONE